# AutoModel: Autonomous Model Development for Image Classification with LLM Agents

## Abstract

Computer vision is a critical component in a wide range of real-world applications, including plant monitoring in agriculture and handwriting classification in digital systems. However, developing high-quality computer vision systems traditionally requires both machine learning (ML) expertise and domain-specific knowledge, making the process labor-intensive, costly, and inaccessible to many. To address these challenges, we introduce AutoModel, an LLM agent framework that autonomously builds and optimizes image classification models. By leveraging the collaboration of specialized LLM agents, AutoModel removes the need for ML practitioners or domain experts for model development, streamlining the process and democratizing image classification. In this work, we evaluate AutoModel across a diverse range of datasets consisting of varying sizes and domains, including standard benchmarks and Kaggle competition datasets, demonstrating that it consistently outperforms zero-shot LLM-generated pipelines and achieves human practitioner-level performance.

## 1 Introduction

Computer vision has emerged as a powerful tool for addressing many complex real-world problems, from plant monitoring in agriculture to handwriting classification in digital systems. However, the process of training computer vision models has grown increasingly complex, involving many different steps such as data augmentation, architecture selection, and hyperparameter tuning. As a result, developing high-performing models is labor-intensive and often demands machine learning (ML) expertise, as well as domain-specific knowledge. Each component must be manually calibrated based on prior experience and detailed analysis of training statistics.

To simplify this process, tools such as Weights & Biases have been developed, offering ML practitioners the ability to track, organize, and optimize model training workflows. Similarly, many approaches in Automated Machine Learning (AutoML), such as hyperparameter optimization (Bergstra & Bengio, 2012; Bergstra et al., 2011; Snoek et al., 2012; Springenberg et al., 2016; Falkner et al., 2018) and neural architecture search (Elsken et al., 2017; Kandasamy et al., 2018), aim to automate parts of the ML pipeline. While these platforms and methods provide valuable support, they tend to address isolated aspects of the model development process and still require substantial involvement from human experts.

Recent advancements in large language models (LLMs) have shown their ability to serve as capable code generators and emulate expert agents in collaborative environments. For instance, Hong et al. (2024) introduced MetaGPT, a multi-agent framework for autonomous software development, where agents assume roles like product managers, software engineers, and quality assurance (QA) engineers. Similarly, Du et al. (2024) proposed Cross-Team Collaboration, where LLM agents work across design, coding, and testing phases in a coordinated multi-team framework.

Inspired by these advances, we propose AutoModel - an LLM agent framework designed to fully automate the model generation process for image classification tasks. AutoModel is an end-to-end framework that requires only a dataset as input. It autonomously handles the entire model development workflow, optimizing various components in parallel, such as data augmentation, architecture selection, and hyperparameter tuning, to produce high-performing models without requiring any user intervention. By assigning specialized roles to LLM agents, AutoModel replicates the collaborative approach of a human expert team, enabling individuals with no ML expertise to train

state-of-the-art image classification models. For experienced ML practitioners, AutoModel can also reduce the time and effort required to develop high-performance models.

Our experiments demonstrate that AutoModel consistently outperforms zero-shot LLM-generated training pipelines and achieves near-human-level performance on standard datasets such as CIFAR-10, TinyImageNet, and various Kaggle competition datasets. This establishes AutoModel as an ideal solution for both domain experts and ML practitioners who seek a streamlined, efficient approach to model development.

Our contributions in this work are as follows

- We present AutoModel, an end-to-end image classification framework that leverages multiple specialized LLM agents to achieve human-level performance in model training.
- We design a fully automated pipeline for code generation, model training, and performance evaluation, which iteratively improves models without any human intervention after initialization.
- We conduct comprehensive experiments on a wide range of datasets, including standard benchmarks and real-world datasets, demonstrating that AutoModel consistently outperforms zero-shot prompting LLMs and matches the performance of expert human practitioners.

## 2 RELATED WORKS

### 2.1 LARGE LANGUAGE MODELS AND LLM AGENTS

Large language models (LLMs) are pre-trained on massive text corpora, often with millions to trillions of parameters. Some of the most well-known LLMs today include GPT-3.5 and GPT-4 (OpenAI et al., 2024) from OpenAI, Claude 3.5 from Anthropic, Mixtral from Mistral, and Llama 3 from Meta. While LLMs can be used directly after pre-training, they often undergo additional training stages, such as supervised fine-tuning (SFT) and reinforcement learning from human feedback (RLHF) (Ouyang et al., 2022). These stages aim to align LLMs with specific behaviors and objectives, resulting in models capable of performing tasks such as general-purpose chatbots, code generation, and information retrieval.

A subset of LLMs specializes in code generation. Codex (Chen et al., 2021), for example, is a GPT model fine-tuned on publicly available GitHub code with a focus on Python programming. Following Codex, Code Llama was introduced, extending coding capabilities to various programming languages such as Python, C++, Java, PHP, and TypeScript. More recently, larger natural language focused LLMs, including GPT-3.5 and Llama 3-8B, have also demonstrated significant coding proficiency across multiple languages, without needing to be explicitly fine-tuned on code.

LLMs have also been used to create human-like agents capable of executing complex instructions autonomously. Examples of such LLM-based agents include ReAct (Yao et al., 2023), Reflexion (Shinn et al., 2023), and SwiftSage Lin et al. (2023), which have demonstrated effectiveness in complex reasoning and decision-making tasks. Expanding on this, works like AutoGPT (Significant Gravitas) enabled LLM agents to move beyond reasoning and autonomously perform actions, such as executing code and receiving feedback from outputs. Another line of research explores the collaboration of multiple LLM agents, each with specialized roles (Hong et al., 2024; Du et al., 2024; Park et al., 2023). These multi-agent frameworks have shown that LLMs can effectively assume distinct roles (Tseng et al., 2024), and role specialization enhances their ability to retrieve relevant knowledge, outperforming single-agent systems on tasks requiring reasoning and strategic planning (Sreedhar & Chilton, 2024).

### 2.2 AUTOMATED MACHINE LEARNING

Automated Machine Learning (AutoML) is a field of research seeking to automate various stages of the machine learning pipeline, making it more efficient to develop ML models. Most AutoML works focus on automating specific parts of the process, such as data preparation, model architecture selection, and hyperparameter optimization (He et al., 2021).

For instance, works like AutoAugment (Cubuk et al., 2019), Faster AutoAugment (Hataya et al., 2019), DADA (Li et al., 2020), and TrivialAugment (Müller & Hutter, 2021) concentrate on automating the data augmentation component of image classification by applying search strategies

over predefined search spaces. Another major focus in AutoML is neural architecture search (NAS), which aims to find the most optimal model architecture from a given search space. A prominent example is Elsken et al. (2017), where a hill-climbing algorithm is used to identify the best Convolutional Neural Network (CNN) architecture. More advanced approaches, such as those by Kandasamy et al. (2018), leverage Bayesian optimization and optimal transport to refine this search.

In addition to NAS, hyperparameter optimization (HPO) is a closely related area that seeks to identify the best set of hyperparameters for a fixed architecture. Random search (Bergstra & Bengio, 2012) is a simple yet efficient method that outperforms traditional grid search (Bergstra et al., 2011) by exploring the search space more effectively. To further enhance the search process, many works incorporate Bayesian optimization in HPO (Snoek et al., 2012; Springenberg et al., 2016; Falkner et al., 2018), enabling better use of past information to guide future decisions.

### 2.3 AutoML with LLMs

With the recent advancement of LLMs, many researchers have started exploring using LLMs to tackle problem in AutoML, as LLMs offer much greater flexibility and in the search space over traditional methods. One of the earliest works in this direction is Yu et al. (2023), which utilizes a fine-tuned Generative Pre-Trained Transformer (GPT) model to create new architectures using crossover, mutation, and selection strategies. Later, many works took advantage of the pre-training of LLMs, which grants them an immense amount of implicit knowledge on neural architectures. GENIUS (Zheng et al., 2023) is one such work, where NAS is performed by simple prompting GPT-4 to generate different configurations, which are evaluated iteratively for a pre-determined number of iterations. In a different direction, Hollmann et al. (2024) proposed the CAAFE method that uses LLMs to automate feature engineering by generating semantically meaningful features for tabular dataset. Zhang et al. (2023) worked on a more comprehensive pipeline, attempting to simultaneously tackling many tasks including object detection, object classification and question answering with LLM-based AutoML. Notably, not only did they use LLMs to improve data augmentation and model architecture components, they also did so based on a LLM-predicted training log, although there were no experimental comparison that demonstrates the method's effectiveness.

## 3 Method

### 3.1 Background

In image classification tasks, the engineering workflow typically begins with a high-level assessment of the dataset. This initial step focuses on understanding the size and complexity of the dataset, ensuring that the subsequent steps are aligned with the specific characteristics of the data.

Next, the workflow moves to data processing, a critical phase that prepares the raw data for model training. This involves standardizing the input images, resizing them to consistent dimensions, and possibly augmenting the dataset through techniques such as rotation, flipping, or color adjustments. These pre-processing steps are essential to ensure that the model can generalize well across different variations of the input data.

Once the data is pre-processed, the focus shifts to model selection and engineering. Engineers consider various architectures and select the one with the most potential to achieve good performance in the context of the given dataset. This phase may also involve incorporating regularization techniques, such as Dropout (Srivastava et al., 2014), as necessary to better suit the specific characteristics of the dataset.

Following model selection, the training process is initiated. This involves configuring the training hyperparameters, such as learning rate, batch size, and optimization algorithms.

After the model is trained, it is evaluated on a test set to measure its performance. Based on the evaluation results, engineers may revisit earlier stages of the workflow - adjusting data processing techniques, exploring alternative model architectures, or refining training parameters - to iteratively improve the model's accuracy and robustness. This cyclical process continues until the model achieves satisfactory performance, ensuring that the final output is well-suited for the object classification task at hand.

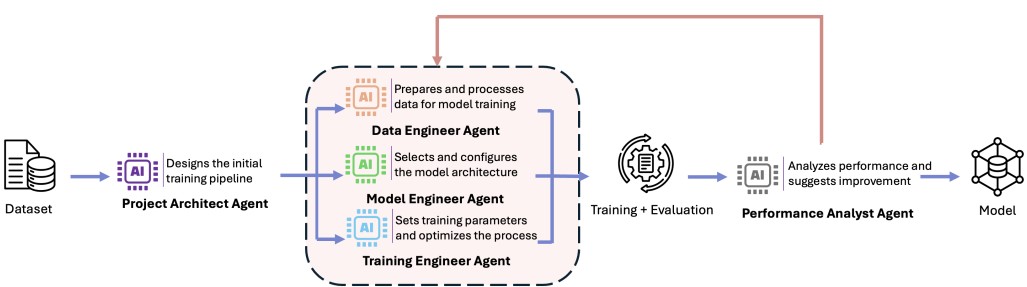

Figure 1: The architecture of the AutoModel Framework

While effective, this traditional workflow is labor-intensive, requiring constant supervision from ML experts at every stage of development. These ML experts must consider the training performance and manually adjust components like data augmentation, model selection, and hyperparameter tuning, which poses a significant barrier to non-experts and limits the scalability of model development.

## 3.2 AUTOMODEL

To address the challenges with traditional model tuning, we wish to design and develop an automate system that simulates the above process with a team of LLM agents. Thus, we introduce AutoModel, an end-to-end framework that utilizes a team of specialized Large Language Model (LLM) agents to autonomously generate a high-performing model, requiring only a dataset as input and no human intervention. We provide a diagram of the workflow of Automodel in 1.

The framework is designed to streamline the machine learning pipeline by assigning specific roles to each agent, ensuring every aspect of model development is handled efficiently and effectively. Each agent is provided with a detailed profile that includes the overall project goal, team structure, individual responsibilities, and instructions for collaboration. To further enhance their effectiveness, each agent is embedded with explicit prompts that guide them in applying advanced techniques and methods that may not be intuitively utilized by LLMs without direct prompting, despite their implicit knowledge of these techniques.

The key agents in the AutoModel framework are as follows:

- **Project Architect**: This agent is responsible for the initial analysis of the dataset, evaluating its size, structure, and any supplementary information provided. Based on this analysis, the Project Architect generates a comprehensive technical design that outlines the data processing steps, model architecture, and training strategies. This design serves as the foundation for the entire workflow.
- **Data Engineer**: The Data Engineer handles the implementation of the data processing pipeline according to the specifications laid out by the Project Architect. This involves tasks such as data standardization, augmentation, and transformation, ensuring that the dataset is optimally prepared for model training.
- **Model Engineer**: The Model Engineer is tasked with selecting and configuring the model architecture. This agent chooses the most appropriate model, applies necessary modifications, and incorporates techniques such as regularization to maximize performance.
- **Training Engineer**: This agent is responsible for configuring the model training process. Key tasks include setting the hyperparameters, such as learning rate and batch size, and selecting optimization algorithms to ensure efficient training.
- **Performance Analyst**: After the model is trained, the Performance Analyst reviews the entire history of pipeline configurations and training logs. This agent identifies areas for improvement and provides targeted feedback to the relevant engineering agent, guiding the next iteration of model development.

Each agent in the AutoModel framework is equipped with knowledge of its role, the team's goals, and how to collaborate effectively. The prompts provided to each agent embed expert-level knowledge specific to their domain. For example, the Model Engineer is aware of effective pre-trained

models like ConvNeXt (Liu et al., 2022), while the Data Engineer is familiar with advanced data augmentation techniques such as MixUp (Zhang et al., 2018) and CutMix (Yun et al., 2019). Although LLMs like GPT-4o implicitly understand these techniques, they rarely apply them unless explicitly prompted. By specifying these techniques in the prompts, AutoModel ensures that the agents select and experiment with highly effective methods when generating the training pipeline.

These specialized agents collaborate to develop and improve models iteratively. The development process begins with the **Project Architect**, who reviews the dataset and additional context to produce a technical design. This design is then passed to the **Data Engineer**, **Model Engineer**, and **Training Engineer**, who collaboratively build their respective components according to the plan.

Once the data pipeline, model architecture, and training configurations are in place, the system integrates these components and trains the model. The **Performance Analyst** then reviews all past and current pipeline configurations, along with their associated training logs, to identify areas for improvement. It provides targeted feedback to the most relevant engineering agent, who adjusts their component accordingly. The updated system is then retrained and re-evaluated.

This iterative cycle continues, with each round focusing on refining the model based on the feedback loop. In particular, configurations that lead to performance gains are saved, while those that degrade performance will be discarded. The process runs for a predefined number of iterations, after which the best-performing configuration and its corresponding model checkpoint are selected as the final output.

AutoModel optimizes multiple aspects of the training pipeline, including data augmentation, optimization strategies, and hyperparameters, all in a single iterative process. Unlike traditional AutoML methods such as hyperparameter optimization and neural architecture search, which focus on specific components, AutoModel's flexibility allows it to explore a wider search space by adjusting nearly all components within the training pipeline. Additionally, AutoModel optimizes the components sequentially rather than simultaneously. This stepwise approach is essential because many components, such as learning rate and batch size, are interdependent. Adjusting these parameters together without considering their interactions can result in suboptimal performance. By focusing on one component at a time, AutoModel effectively identifies and builds upon successful changes, leading to more reliable improvements in future iterations.

### 3.3 IMPLEMENTATION

With the architecture and roles of the AutoModel framework introduced, we now discuss its implementation. In particular, we will discuss how the framework processes dataset inputs, autonomously generates and executes code, and employs iterative refinements to optimize model performance.

**Dataset Information Input** A key advantage of the LLM agents framework is its ability to understand any natural language description of the dataset, enabling the system to make more informed decisions. Optionally, the user can supply a brief description of the dataset, detailing aspects such as the image domain (e.g., real-life objects, digits, synthetic images), whether the images are grayscale or colored, their dimensions (same or varying sizes), and their quality (high or low). This information is passed to both the Project Architect and Performance Analyst and is factored into the initial technical design and performance analysis. The impact of including this dataset information is discussed further in Section 4.4.

**Autonomous Code Execution** The AutoModel framework autonomously generates, executes, and refines code over multiple iterations to improve model performance. The code-generating agents (i.e., Data Engineer, Model Engineer, and Training Engineer) operate according to a pre-defined interface. This interface specifies the required input arguments and expected return values for each component. The agents have flexibility in how they implement their code, but they must adhere to these interface guidelines. Once the agents produce the code, it is parsed, extracted, and saved into Python files. The system then uses a pre-defined pipeline to integrate these components and execute the full model training process. The execution outputs, including model performance metrics, are captured at the shell level and stored in a training log for analysis.

**Zero-shot Initialization** The Project Architect agent leverages the zero-shot generation capabilities of LLMs to produce a strong baseline training pipeline, ensuring a foundation of good performance from the outset. As discussed in the ablation studies in Section 4.3, generating the data, model,

and training components sequentially in separate calls leads to suboptimal results, even when the LLM has access to previously generated components. To address this issue, the Project Architect generates a single, coherent pipeline in one step, which is then divided into the necessary components (data, model, and training) required by AutoModel. This approach allows the framework to be more efficient by starting from a solid baseline, reducing the need for extensive corrections and minimizing wasted iterations on poor initial configurations.

**Performance Analysis with Summarized History** To ensure informed decision-making and prevent redundancy, all previous pipeline configurations (e.g., data augmentation methods, model architectures, hyperparameters) and their corresponding training logs are passed to the Performance Analyst. However, passing the entire history of code and logs could quickly exceed the LLM's maximum context length. To address this, after each successful iteration, the system uses an LLM to summarize the key aspects of the data, model, and training code, highlighting critical details such as the methods and hyperparameters used. The summarized configuration history, along with the corresponding training performance data, is then provided to the Performance Analyst to guide further iterations and improvements.

# 4 EXPERIMENTS

## 4.1 EXPERIMENTAL SETTING

**Datasets** We evaluated AutoModel using two widely recognized object classification datasets: CIFAR-10 (Krizhevsky & Hinton, 2009) and TinyImageNet (Deng et al., 2009). CIFAR-10 contains 60,000 32x32 color images across 10 distinct classes, while TinyImageNet, a subset of ImageNet1K, includes 100,000 images across 200 classes, resized to 64x64 pixels. Both datasets feature commonly seen objects such as vehicles and animals.

To test AutoModel's robustness, we also incorporated CIFAR-10-C (Hendrycks & Dietterich, 2019), a variant of CIFAR-10 designed to assess model performance under common corruptions. CIFAR-10-C introduces 15 corruption types, grouped into four categories: noise, blur, weather, and digital distortions. These corruptions are applied at two severity levels (1 and 5), resulting in 30 distinct versions of the dataset. For experimental efficiency, we created two subsets, CIFAR-10-C-1 and CIFAR-10-C-5, where the number indicates the severity level. These subsets were generated by uniformly sampling one image from each corrupted dataset for every test image.

Additionally, we evaluated AutoModel on two smaller datasets from different domains: SVHN and dSprites. SVHN is a digit classification dataset containing real-world images of house numbers, with variations in resolution and background complexity. dSprites is a synthetic dataset of 2D shapes, with latent factors controlling the shapes' properties. Specifically, we used the dSprites Orientation dataset, where each shape is rotated into one of 40 possible orientations within the $[0, 2\pi]$ range. Both datasets are part of the Visual Task Adaptation Benchmark (VTAB) (Zhai et al., 2020).

To further assess AutoModel's versatility across different real-world domains, we also evaluated it on four diverse datasets from Kaggle, a leading platform for machine learning competitions. These datasets span a range of sizes and represent various data domains, including:

- **Cassava Leaf Disease Classification** (Mwebaze et al., 2020): This dataset consists of 21,367 photos of cassava plant in Uganda, mostly taken using inexpensive cameras. The model must identify whether the plant has one of the four diseases: Mosaic Disease, Green Mottle Brown Streak Disease, Bacterial Blight Disease, or no disease, with 5 classes in total.
- **Kitchenware Classification** (ololo, 2022): This dataset has 9,367 photos of 6 types of common kitchenware, including cups, glasses, plates, spoons, forks and knives. These photos are taken in households with various lighting and scene conditions.
- **Arabic Letters Classification** (Khalil, 2023): This dataset contains 53,199 images of 65 written Arabic letters, each exhibiting positional variations based on their occurrence within a word with four possible forms: isolated, initial, medial and final. These letters were collected from 82 different users.
- **4 Animal Classification** (Lee et al., 2022): This dataset comprises a total of 3,529 images, categorized into four distinct animal classes: cats, deer, dogs, and horses. Each class represents a diverse range of images capturing various poses, environments, and lighting conditions.

Table 1: Average classification accuracy of models generated by AutoModel and zero-shot prompting LLMs on standard datasets from three trials. We report the accuracy after the first iteration, as well as the final accuracy after 20 iterations to show improvement. The best accuracy on each dataset is bolded.

| Dataset | LLM Zero-shot | AutoModel First Iter. | AutoModel Final Acc. |
|---|---|---|---|
| CIFAR-10 | 0.9290 | 0.9171 | **0.9533** |
| CIFAR-10-C-1 | 0.5425 | 0.9121 | **0.9476** |
| CIFAR-10-C-5 | 0.6218 | 0.9359 | **0.9422** |
| TinyImageNet | 0.4815 | 0.4225 | **0.7875** |

Three key criteria guided our selection of Kaggle datasets to ensure a streamlined and fair evaluation. First, we excluded datasets with more than a million images to maintain computational feasibility. Second, we prioritized datasets from public competitions, allowing for direct comparison between AutoModel's performance and that of human ML practitioners. Third, we focused on competitions that used top-1 accuracy as the primary evaluation metric, avoiding those that relied on metrics such as the area under the ROC curve, F1 score, or multiclass log loss. After applying these filters, these four datasets were among the few that met all the criteria and aligned with the goals of our evaluation.

For all datasets, we use the provided test set when available. If no test set is provided, we designate the validation set as the test set. In cases where only a single unsplit dataset is available, we manually split the dataset into training and test sets using an 80-20 ratio. This ensures consistency across all experiments while maintaining a fair evaluation process.

**Baseline** For VTAB datasets, namely SVHN and dSprites Orientation, we compare our results with that of Visual Prompt Tuning (VPT) (Jia et al., 2022), a well-regarded method in the domain of fine-tuning, since AutoModel usually chooses to fine-tune based on a pre-trained model. For all four Kaggle datasets, we compare with the existing submissions on the leaderboard of the competition. As one of the first works in the domain of fully automating model generation, there is little existing work baseline we can compare to. Thus, to provide additional comparison, we choose to compare the accuracy from models generated by AutoModel with the likely approach of a non-ML experts: asking LLMs to directly generate a training script with zero-shot prompting. For this, we directly ask the LLM to "generate code for training a model on a dataset with x classes." We run the code to train a model, and evaluate the model on the test set.

**Experimental Setup** To ensure consistency, we used GPT-4o, a commercial LLM developed by OpenAI, for all experiments. The experiments weres conducted over three trials, with the average accuracy reported. For all AutoModel runs, we performed 20 iterations to balance optimization and experimental efficiency. For the VTAB datasets (SVHN and dSprites Orientation), we specifically instructed the Model Engineer to use the Vision Transformer (ViT), specifically the ViT-B/16 model, to ensure a fair comparison with the VPT paper, which used the same model.

## 4.2 MAIN RESULTS

The results in Table 1 demonstrate that AutoModel consistently outperforms models generated by zero-shot prompting LLMs, confirming its effectiveness as a superior alternative for non-ML experts seeking to train high-performing models. For simpler datasets like CIFAR-10, AutoModel achieves a slightly higher accuracy (95.33%) compared to zero-shot prompting (92.90%). However, its strength becomes more evident on challenging datasets like TinyImageNet, where AutoModel reaches an impressive 78.75% accuracy, nearly 31% higher than the zero-shot baseline of 48.15%. Additionally, AutoModel shows strong performance on the robustness datasets, namely CIFAR-10-C-1 and CIFAR-10-C-5, highlighting its capability to generate models that are both high-performing and robust under varying conditions.

Table 2 further illustrates AutoModel's capabilities by comparing it to Visual Prompt Tuning (VPT) (Jia et al., 2022), a well-regarded fine-tuning method. AutoModel not only surpasses VPT but also outperforms the full-parameter fine-tuning baselines reported by the original work. On both

datasets, SVHN and dSprites Orientation, AutoModel shows clear improvements over zero-shot prompting too.

Table 2: Average classification accuracy of models generated by AutoModel and zero-shot prompting LLMs on two VTAB datasets: SVHN and dSprites Orientation. We report the accuracy after the first iteration, as well as the final accuracy after 20 iterations to show improvement. Full-parameter fine-tuning and visual prompt tuning (VPT) results from Jia et al. (2022) are both included for comparison. The best accuracy on each dataset is bolded.

| Dataset | LLM Zero-shot | Full Fine-tuning | VPT | AutoModel First Iter. | AutoModel Final Acc. |
|---|---|---|---|---|---|
| SVHN | 0.9250 | 0.8740 | 0.7810 | 0.9419 | **0.9695** |
| dSprites Orientation | 0.6515 | 0.4670 | 0.4790 | 0.9150 | **0.9522** |

In addition to standard datasets, AutoModel's performance on non-standard Kaggle datasets is presented in Table 3. These datasets vary in size, image quality, and domain, providing a more realistic test of AutoModel's adaptability in the real world. Across all four datasets, AutoModel consistently achieves better results than zero-shot prompting. While AutoModel's top-1 accuracy slightly lags behind the best Kaggle leaderboard results, this is expected since AutoModel was set to run for only 20 iterations. Approximately half of these iterations encountered code issues that caused the code to throw an error midway, such as undefined variables, package misuses, or tensor shape errors. As a result, only around 10 iterations were fully executed and analyzed, limiting the opportunity for refinement. This turns out to be comparable to the number of attempts made by human practitioners in real-world competitions. Note that human code on Kaggle is often extensively tuned and error-free before submission. Thus, AutoModel can be said to perform nearly on par with top human ML practitioners in these Kaggle competitions, showcasing its potential as a powerful tool for model development even under real-world constraints.

Table 3: Average classification accuracy of models generated by AutoModel and zero-shot prompting LLMs on four Kaggle datasets. We report the accuracy after the first iteration, as well as the final accuracy after 20 iterations to show improvement. AutoModel's competition rank, top accuracy on Kaggle, and the average number of submission attempts in the top 5 positions are also reported.

| Dataset | LLM Zero-shot | AutoModel First Iter. | AutoModel Final Acc. | Kaggle Statistics | | |
|---|---|---|---|---|---|---|
| | | | | Rank | Top Acc. | Top Attempts |
| Cassava Leaf Disease | 0.7748 | 0.7493 | **0.8574** | 2892/3900 | 0.9152 | 98 |
| Kitchenware | 0.8581 | 0.9475 | **0.9793** | 25/115 | 0.9991 | 12 |
| Arabic Letters | 0.5946 | 0.8212 | **0.8403** | 85/177 | 0.9680 | 10 |
| 4 Animals | 0.9196 | 0.8934 | **0.9518** | 184/221 | 0.9958 | 10 |

## 4.3 ABLATION STUDIES

**Zero-shot Initialization**   As described earlier, the Project Architect generates a complete training pipeline in a single call using zero-shot prompting, which is then broken down into multiple components. This strategy ensures a strong initialization configuration for the model. In contrast, generating each component sequentially - first generating data augmentation code, passing it to the Model Engineer, and then passing the data and model code to the Training Engineer - leads to less coherent code and decreased model performance.

In Table 4, we compare the performance of AutoModel with and without zero-shot initialization. Even though AutoModel without zero-shot initialization can still improve iteratively and outperform the baseline LLM-generated model, the final accuracy is notably lower than that achieved using zero-shot initialization. This highlights the importance of a strong initial configuration, which enables AutoModel to converge to a high-performing model more efficiently.

**Smaller LLMs**   In our experiments, we primarily utilized GPT-4o, a high-performing large language model. However, to evaluate AutoModel's robustness with smaller models, we also tested its performance using GPT-4o-mini. GPT-4o-mini is designed as the official successor to GPT-3.5,

Table 4: Classification accuracy of models generated by AutoModel (with and without zero-shot initialization) and zero-shot LLMs on CIFAR-10-C-1 and CIFAR-10-5.

| Dataset | No Zero-shot Initialization | AutoModel | LLM Zero-shot |
|---|---|---|---|
| CIFAR-10-C-1 | 0.8297 | 0.9476 | 0.5425 |
| CIFAR-10-C-5 | 0.7942 | 0.9422 | 0.6218 |

offering improvements in cost, speed, and computational efficiency while maintaining strong performance.

As shown in Table 5, AutoModel continues to deliver strong results even when using GPT-4o-mini. Despite its smaller size, the model achieves classification accuracies that are significantly higher than those obtained by zero-shot LLMs. This demonstrates AutoModel's ability to perform well even with limited budget.

Table 5: Average classification accuracy of models generated by AutoModel and zero-shot LLMs on CIFAR-10-C-1 and CIFAR-10-5, using GPT-4o-mini.

| Dataset | AutoModel | LLM Zero-shot |
|---|---|---|
| CIFAR-10-C-1 | 0.9557 | 0.6423 |
| CIFAR-10-C-5 | 0.8607 | 0.6503 |

## 4.4 UTILIZING DATASET INFORMATION

In our experiments, we investigated how providing additional dataset-specific information can influence the effectiveness of data augmentation strategies in AutoModel. When supplied with this information, AutoModel is capable of selecting augmentations that are well-suited to the characteristics of the dataset.

For instance, when training on the SVHN dataset, which consists of real-world images of house numbers, AutoModel chose to apply the ColorJitter augmentation. The reasoning provided was that "Color jitter (brightness, contrast, saturation, hue) can help in robustifying the model against variations in lighting conditions." This augmentation is particularly appropriate for SVHN since real-world images can be subject to inconsistent lighting, and this technique helps improve model robustness in such conditions.

However, not all augmentations are universally beneficial. In some cases, certain augmentations can harm model performance. For example, in the dSprites Orientation dataset, where orientation is label for classification, applying any flipping augmentation like RandomHorizontalFlip can degrade performance by altering the class labels and confusing the model. AutoModel's Performance Analyst identified this issue, noting: "RandomHorizontalFlip may not be useful for orientation classification tasks," and "Orientation-based tasks will not benefit from horizontal flipping; could even confuse the model."

This demonstrates AutoModel's ability to intelligently adapt its augmentation strategy based on the specific characteristics of the dataset. By recognizing which augmentations enhance performance and which are detrimental, AutoModel fine-tunes the model more effectively, leading to better and faster optimization.

## 5 CONCLUSION

In this paper, we introduced AutoModel, an end-to-end LLM agent framework designed to autonomously generate high-performing image classification models. As a part of the framework, we designed an automated code generation and execution pipeline, eliminating the need for human intervention, requiring only a dataset as input. By leveraging the expertise of specialized agents, AutoModel replicates the workflow of human practitioners, to iteratively improve on tasks such as data augmentation, model selection, and hyperparameter tuning.

Our experiments demonstrated that AutoModel consistently outperforms models generated by zero-shot LLM prompting. AutoModel is capable of handling both corrupted datasets and adapt to varying domains, showcased through its evaluation on Kaggle competition datasets, where it performed comparably to top human ML practitioners.

In summary, AutoModel represents a significant advancement in automating model development for image classification. Its ability to perform on par with expert human practitioners while being fully autonomous demonstrates its potential to democratize access to machine learning and streamline model development for both experts and non-experts alike.

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
