# OpenReview forum: "AutoModel: Autonomous Model Development for Image Classification with LLM Agents"
_ICLR.cc/2025/Conference — ICLR 2025 Conference Withdrawn Submission_

### Official Review · Reviewer_VTEu · 2024-10-22

**Soundness:** 3
**Presentation:** 3
**Contribution:** 2
**Rating:** 3
**Confidence:** 4

**Summary:**

The paper presents "AutoModel", a new LLM agent framework that can automatically build and optimize a vision model for image classification. Specifically, AutoModel utilizes several specialized LLM agents for designing the training pipeline, processing the training data, configuring the training model, setting the training hyper-parameters, and analyzing the performance. By taking only a dataset as the input, AutoModel is an end-to-end AutoML framework. Extensive experiments on image classification tasks validate the effectiveness of the proposed AutoModel.

**Strengths:**

1. The proposed LLM agent framework for end-to-end AutoML is new.
2. AutoModel is an end-to-end framework with only a dataset as the input.
3. Compared with LLM-generated models, the proposed method achieves better performance.

**Weaknesses:**

1. Lack of novelty. I think the proposed AutoModel is more like a specific application of MetaGPT in AutoML. In MetaGPT, LLM agents are given different prompts to be different specialized experts, and will together finish a programming project. In AutoModel, LLM agents will act in different roles with different prompts given and participate in formulating an ML pipeline, which is one special programming project that MetaGPT can also finish. Besides, the essential idea of utilizing LLM as different specialized experts to collaborate on a project is almost the same. It will be better for the authors to highlight AutoModel's advantages over MetaGPT in AutoML. To further improve the presentation of AutoModel, I think the authors can analyze the differences in the mechanism of AutoML cooperation between AutoModel and MetaGPT, and explain why AutoModel is better for usage.
2. The authors didn’t compare the model scale or the computational cost of the generated models in experiments, which makes the improvement of the performance not meaningful. The model engineer agent can always output a larger model for training to achieve better performance, without any constraints of the model scale or the computational cost. To address this concern, I suggest that the authors give prompts with model scale constraints (e.g., less than 20M) to relevant agents and report the performance with the corresponding model scale for a fair and significant comparison.
3. There are several full-pipeline AutoML methods that the authors didn’t compare or mention [1][2]. It will be better for the authors to analyze the differences or the improvement of AutoModel over these works.
4. The authors only explored image classification experiments on small-scale datasets (CIFAR/Tiny-ImageNet/…). It will be better for the authors to explain the reason for choosing these datasets instead of large-scale ones (ImageNet-1K). When scaling up to large-scale datasets, what challenges will AutoModel face? This item will not affect my ratings.

[1] Lanqing, H. O. N. G., et al. "DHA: End-to-End Joint Optimization of Data Augmentation Policy, Hyper-parameter and Architecture." Transactions on Machine Learning Research (2022).

[2] Wang, Zhaozhi, et al. "Multi-Agent Automated Machine Learning." Proceedings of the IEEE/CVF Conference on Computer Vision and Pattern Recognition. 2023.

**Questions:**

Please refer to weaknesses. My main concern is that the contributions lack novelty, and I will raise my ratings if the authors' response can address it well. Besides, the experiments are not really meaningful, as I've claimed in weaknesses (2).

---

### Official Review · Reviewer_k1jr · 2024-10-28

**Soundness:** 2
**Presentation:** 2
**Contribution:** 2
**Rating:** 3
**Confidence:** 1

**Summary:**

The paper proposes a general framework to automatically generate a neural network for classification tasks by LLM. This can be a good alternative for traditional NAS for architecture search, AutoAugment for data augmentation etc that only specialize on a specific step for network design. Figure 1 outlines the whole framework. It uses LLM agent for each component in traditional machine learning pipeline design. Experiments show that the proposed framework can outperform LLM Zero-shot and VPT.

**Strengths:**

The objective of the paper is clear. It aims to provide ML practitioners an automatic tools for network design.

The topic aligns with ICLR.

**Weaknesses:**

The presentation is poor. Many key details are missing. It is impossible to reproduce any results.

The technicial novelty is limited. It seems like just to prompt GPT-4o and combines its outputs.

**Questions:**

For the compared methods, what does “zero-shot LLM-generated training pipelines” mean?

Now that the AutoModel can select the optimal architecture, why specifically chose ViT-B/16 (line 361).

For each component in section 3.2, what exactly their input and output, how to gather them and bring them together? For example, for data engineer, what exactly the prompt looks like, some training image examples? If so, how to sample them. The statistics of the training set? Anyway, across the whole paper, there are tons of information are missing which caused it unclear for each component of the paper. This makes it impossible to reimplement the results.

What is exactly the best model the AutoModel make? And the best training recipe? Are they the same for each dataset? If not, why?

The literature review discussed AutoAugment, NAS, etc methods but no comparison in experiments.

For each component, the LLMs used are the same GPT-4o or mini, both are general LLM. Shouldn’t specialist LLM agent for each component?

---

### Official Review · Reviewer_7Eid · 2024-10-30

**Soundness:** 3
**Presentation:** 2
**Contribution:** 2
**Rating:** 3
**Confidence:** 4

**Summary:**

This paper presents an LLM agent framework for AutoML, especially for image classification models. The framework consists of Project Architect, Data Engineer, Model Engineer, Training Engineer, and Performance Analyst. Experiments are conducted on a diverse range of benchmark datasets and Kaggle competition datasets.

**Strengths:**

1. The paper is well-motivated. Utilizing LLMs for AutoML is a promising direction.

2. The design choice of different modules are reasonable.

3. The experiments are conducted on a diverse range of datasets consisting of varying sizes and domains.

**Weaknesses:**

1. Lack of comparisons against some related works [a,b,c]. The paper does neither discuss their difference nor compare with them.

2. The baseline methods are not strong enough. Only zero-shot prompting LLMs are compared. More experiments are required.
Please compare with traditional AutoML methods (e.g. HPO, AutoAugment etc).

3. The performance of the proposed method is not good enough, as the ranking on Kaggle is not high enough (ranking 2892/3900). It argues that the performance can be significantly improved after multiple rounds of optimization, but there are no examples or analyses of the reasons for this improvement, nor is there a demonstration of the process over 20 rounds. Furthermore, the model after 20 optimizations still ranks low on Kaggle.

4. Lack of ablation study of different modules (e.g. different agents).

5. Limited scope. The paper only conducts experiments on image classification, ignoring more comprehensive tasks, e.g. object detection, image segmentation (these tasks are commonly evaluated in previous related works [a]).

6. The description and framework diagram lack details about the agents and their collaboration processes. More detailed examples and prompts for these agents are required for reproduction.


[a] Yang Z, Zeng W, Jin S, et al. AutoMMLab: Automatically Generating Deployable Models from Language Instructions for Computer Vision Tasks[J]. arXiv preprint arXiv:2402.15351, 2024.

[b] Viswanathan V, Zhao C, Bertsch A, et al. Prompt2model: Generating deployable models from natural language instructions[J]. arXiv preprint arXiv:2308.12261, 2023.

[c] Zhang S, Gong C, Wu L, et al. Automl-gpt: Automatic machine learning with gpt[J]. arXiv preprint arXiv:2305.02499, 2023.

**Questions:**

1. Details about the agents and their collaboration processes. And why these designs are novel?

---

### Official Review · Reviewer_a9s7 · 2024-11-03

**Soundness:** 2
**Presentation:** 3
**Contribution:** 2
**Rating:** 3
**Confidence:** 4

**Summary:**

The paper presents a framework designed to autonomously develop and optimize image classification models using large language model (LLM) agents. Inspired by multi-agent collaborative frameworks, AutoModel assigns roles to specialized LLM agents that collaboratively handle each stage of the model development pipeline—from data processing to model training and evaluation—without requiring human intervention. The authors motivate this framework with the potential to facilitate the setup of image classification models in real-world scenarios without domain knowledge. Further, they claim that their experiments demonstrate that AutoModel achieves human-like performance across several standard and real-world datasets, comparing them to Kaggle benchmarks. They also demonstrate the effectiveness of their iterative method by showing higher classification accuracies compared to zero-shot LLM-generated training pipelines.

**Strengths:**

1. Framing the development of an image classification problem into a LLM agent framework is an interesting idea and the choice of agents/components needed for an automated end-to-end framework seems reasonable and well thought through. In my opinion, the main strength of this approach is that it can take dataset-specific information into account to optimise the different system components.
2. The choice of datasets seems fair as both standard benchmark and also non-standard datasets are used.
3. Experiments suggest that the proposed iterative approach indeed leads to consistent model improvement.

**Weaknesses:**

1. The practical impact of this work is limited. Establishing an image classification model for a practical use case is not challenging these days even for non-experts. There are several low-effort approaches that can be used to simplify the implementation. The authors mention AutoML. Other simple classification approaches that do not require extensive model tuning are for example CLIP zero-shot classification; or using a pre-trained foundation model (e.g. DINO) as a feature generator and fit a simple linear or kNN classifier on top of it. Previous work has shown that these approaches can lead to sufficient accuracy in many real-world applications while being much simpler to implement than the approach presented in this paper.
2. Benchmarking the results against other AutoML frameworks seems to be essential for this paper but is missing. In addition, I would recommend to add comparisons to other low-effort approaches as described in the previous point.
3. There are no experiments focusing on the importance of having different agents for the subtasks. Having an analysis that shows which agents contributed most to model improvement would be insightful. Also, it is not clearly demonstrated that the multi-agent setup is superior to an iterative single-LLM setup.
4. Accuracies are reported without errors. Experiments should be repeated multiple times to assess the robustness of results.
5. Experiments are mainly conducted using a single LLM (GPT-4o). The ablation study on smaller LLMs seems insufficient. Showing the effect of LLM choice on different metrics such as the overall accuracy or the rate of erroneous code produced would be insightful.
6. Section 4.4 is meant to address how the framework makes use of dataset-specific information. In my opinion, this could be the core strength of this approach. However, the section feels insufficient as it provides limited anecdotal evidence which does not convince me that AutoModel “intelligently” adapts to dataset information rather than improving the classification model by random chance. I would recommend to research this aspect further by conducting structured experimentation. If you can show that your method uses dataset information that other AutoML approaches are not able to use, this would be a strong finding.
7. Supplementary material covering implementation details (e.g. specific prompts for the agents) is missing and would help understanding the presented approach.

Overall the paper seems somewhat incomplete in its current state. However, I encourage the authors to continue working on their approach is it has the potential to generate insightful results if the right experiments are conducted.

**Questions:**

1. Why did you choose to only report AutoModel’s accuracy after the first and final iterations? A curve would have been useful to assess how stable the improvement is over iterations.
2. In L. 399ff you claim that the code error-rate is comparable to humans, how do you come to that conclusion?
3. Did you limit the LLM to only use certain programming languages or frameworks?
4. Did you automate the step from code generation to code execution? If yes, how did you realise this in practice?

**Minor comments:**

- Line 186: Missing “Figure” in cross-reference.
- Lines 355, 458, 466, 467: Use of non-english quotation marks.

---

### Author Response · Authors · 2024-11-13
**Thank you for the review**

We sincerely thank the reviewers for their thoughtful and thorough feedback. After careful consideration, we have decided to withdraw the paper in order to take the necessary time to address the highlighted weaknesses and make improvements to the paper.

---

### Note · Authors · 2024-11-13

I have read and agree with the venue's withdrawal policy on behalf of myself and my co-authors.